# (20S) Ginsenoside Rh2 Inhibits STAT3/VEGF Signaling by Targeting Annexin A2

**DOI:** 10.3390/ijms22179289

**Published:** 2021-08-27

**Authors:** Yu-Shi Wang, Chen Chen, Shi-Yin Zhang, Yang Li, Ying-Hua Jin

**Affiliations:** Key Laboratory for Molecular Enzymology and Engineering of the Ministry of Education, School of Life Sciences, Jilin University, Changchun 130012, China; wangyushi0317@hotmail.com (Y.-S.W.); cchen16@mails.jlu.edu.cn (C.C.); zsyshiyin@163.com (S.-Y.Z.); liyang915@jlu.edu.cn (Y.L.)

**Keywords:** STAT3, Annexin A2, (20S)G-Rh2, VEGF

## Abstract

Signal transducers and activators of transcription 3 (STAT3) acts as a transcriptional signal transducer, converting cytokine stimulation into specific gene expression. In tumor cells, aberrant activation of the tyrosine kinase pathway leads to excessive and continuous activation of STAT3, which provides further signals for tumor cell growth and surrounding angiogenesis. In this process, the tumor-associated protein Annexin A2 interacts with STAT3 and promotes Tyr705 phosphorylation and STAT3 transcriptional activation. In this study, we found that (20S) ginsenoside Rh2 (G-Rh2), a natural compound inhibitor of Annexin A2, inhibited STAT3 activity in HepG2 cells. (20S) G-Rh2 interfered with the interaction between Annexin A2 and STAT3, and inhibited Tyr705 phosphorylation and subsequent transcriptional activity. The inhibitory activity of STAT3 leaded to the negative regulation of the four VEGFs, which significantly reduced the enhanced growth and migration ability of HUVECs in co-culture system. In addition, (20S)G-Rh2 failed to inhibit STAT3 activity in cells overexpressing (20S)G-Rh2 binding-deficient Annexin A2-K301A mutant, further proving Annexin A2-mediated inhibition of STAT3 by (20S)G-Rh2. These results indicate that (20S)G-Rh2 is a potent inhibitor of STAT3, predicting the potential activity of (20S)G-Rh2 in targeted therapy applications.

## 1. Introduction

STAT3, a member of signal transducers and activators of transcription (STATs) family, participates in various critical cellular processes including cell proliferation, survival, differentiation, and angiogenesis [1,2,3,4,5]. Like other STAT family members, STAT3 is predominantly activated by cytokines like EGF and IL-6 [3,4,5,6,7]. Cytokines activate receptor-linked tyrosine kinases like Src family members and JAKs, which phosphorylates STAT3 and induces the homologous dimerization and nucleus localization, then STAT3 binds to DNA sequences of target genes including cyclins, anti-apoptosis proteins, vascular endothelial growth factors (VEGFs), matrix metalloproteinases (MMPs), and immuno-suppressive proteins [5,8,9].

In normal cells, transient activation of STAT3 produces transcriptional signals from extra-cellular cytokines [1,10,11]. In contract, STAT3 transforms to hyper- and sustaining-activated in cancer for the excessive cytokine supply and ultra-active mutation within kinase pathway, which promotes cancer metastasis, angiogenesis, and immuno-suppression. Moreover, cytokines regulated by STAT3 in tumor microenvironment, like IL-6, IL-10, VEGF and TGF-β, trigger a positive feed-back amplification of STAT3 in multiple types of cells throughout the tumor microenvironment, finally resulting in severer tumor growth and immuno-suppression [1,4,5,8,9,10,11]. Aberrant STAT3 hyper-activation has been estimated to present in over 70% of human cancer and reported in almost all types of malignancies with a poor clinical outcome [5,12,13], and STAT3 together with its relative signaling pathway has long been recognized as a potential therapeutic target [1,5,7,9,12].

Annexin A2 (Anxa2) is well-described as the heavy chain of Anxa2-S100A10 (AⅡt) complex for plasmin processing [14,15]. Recent researches on cancers indicate that Anxa2 is over-expressed in various types of cancers, promotes cancer development, metastasis and chemo-resistance, highly associated with poor prognosis [16,17,18,19,20,21]. Detailed mechanisms are revealed that Anxa2 interacts with transcriptional factors and enhances their activation including NF-κB, STAT3/6 and YAP1 [22,23,24,25,26]. In our previous work, we have already identified Anxa2 as a cellular target of (20S) ginsenoside Rh2((20S)G-Rh2), a natural ginseng extract with anti-cancer activity, and (20S)G-Rh2 inhibited NF-κB activation by interfering Anxa2-NF-κB p50 subunit interaction [27,28]. Here, we will focus on the targeted effect of (20S)G-Rh2 towards Anxa2-STAT3 interaction, and further establish a new therapeutic approach targeting STAT3.

## 2. Results

### 2.1. Anxa2 Regulates VEGF Expression and Secretion

Anxa2 has been indicated as a positive regulator of STAT3 by binding to STAT3 and promoting Tyr705 phosphorylation [23,24]. In order to expound the regulatory effect of Anxa2 on STAT3 activation, we lowered the expression of Anxa2 in HepG2 cells by Anxa2-shRNA, and both Tyr705 phosphorylation and transcriptional activation of STAT3 were inhibited and accorded with the protein level of Anxa2 (Figure 1A,B). Then, the expression of STAT3 target genes were determined, mainly including various cyclins, cell division cycle (CDC) proteins and VEGFs. Cyclins and CDC proteins presented differential shift tendency, while four VEGFs were all down-regulated by Anxa2 expression suppression (Figure 1C), both intracellular and secreted VEGF proteins were lessened as well (Figure 1D,E).

### 2.2. Anxa2 in Tumor Cells Promotes Proliferation and Migration of Co-Cultured HUVECs

VEGFs secreted by tumor cells and stromal cells stimulate proliferation and survival of vascular endothelial cells, promoting possible angiogenesis on both physiological and pathological conditions [29,30,31,32,33]. Three co-culture systems were developed with HUVECs and HepG2 cells to evaluate the effect of secretory VEGFs, and HepG2 cells were pre-treated with Anxa2-shRNA to knock-down Anxa2 expression or not (Figure 2A). The culture medium (CM) of HepG2 cells were isolated and utilized for colony formation of HUVECs. The colony population in HepG2 CM was significantly larger than that of common CM (Figure 2B left), and the colony staining presented the same result that HepG2 CM strengthened the colony formation capability of HUVECs (Figure 2C). The cell cycle of HUVECs was determined by flow cytometry, and co-culture with HepG2 cells decreased G0/G1-phase-cells and increased S/G2/M-phase cells (Appendix A). EdU incorporation was performed for the measurement of newly synthesis DNA and newly proliferating cells, and the ratio of EdU-positive cells was significantly higher in co-cultured HUVECs than in cells cultured alone (Figure 2D,E). An invasive assay was then performed via trans-well method, and HUVECs presented higher invasiveness with HepG2 cells in the low chamber (Figure 2F and Appendix A). Notably, the pro-proliferation and pro-migration capability of HepG2 cells in co-culture systems was largely inhibited by lowering Anxa2 expression by Anxa2-shRNA, and this inhibitory effect was accorded with Anxa2 expression level (Figure 1A,B, Figure 2B–F and Appendix A).

### 2.3. (20S)G-Rh2 Inhibits STAT3 Activation by Targeting Anxa2

Anxa2 has been identified as a cellular target of (20S)G-Rh2 [27,28]. A thermal shift assay was first performed to determine Anxa2-(20S)G-Rh2 interaction. The enhanced thermal stability under (20S)G-Rh2 indicated that both unphosphorylated Anxa2 and pTyr23-Anxa2 interacted with (20S)G-Rh2 (Figure 3A,B). As Anxa2-Tyr23 phosphorylation by c-Src is essential for STAT3 binding and following STAT3 activation [23,24], chemicals modifying Anxa2-Tyr23 phosphorylation were engaged. Saracitinib, as a specific inhibitor inhibited c-Src activity as well as following pTyr23-Anxa2 level, which inhibited Anxa2-STAT3 interaction and STAT3 activation (Figure 3C–E). Src inhibitor 3 inhibited c-Src c-terminal kinase (Csk), a kinase responsible for c-Src kinase-loss phosphorylation, acting as a c-Src activator [34]. Src inhibitor 3 promoted Anxa2-Tyr23 phosphorylation and down-stream STAT3 activation (Figure 3C–E). (20S)G-Rh2 interfered Anxa2-STAT3 interaction in rest state or under stimulus towards c-Src (Figure 3C). The phosphorylation on Tyr705 and activation of STAT3 were then inhibited under (20S)G-Rh2 treatment (Figure 3D,E). Moreover, the constitutively active c-Src (Y530F mutant, Src-CA) was overexpressed for the excessive Anxa2-Tyr23 phosphorylation. Anxa2-Tyr23 phosphorylation and Anxa2-STAT3 interaction were enhanced by Src-CA, and following STAT3-Tyr705 phosphorylation and transcription activity were then enhanced (Figure 3F–H). (20S)G-Rh2 showed no inhibitory effect on Anxa2-Tyr23 phosphorylation, nevertheless inhibited Anxa2-STAT3 interaction, STAT3-Tyr705 phosphorylation and STAT3 activation (Figure 3F–H).

### 2.4. (20S)G-Rh2 Inhibits Proliferation and Migration of HUVECs

The expression of STAT3 target genes in HepG2 cells was determined under (20S)G-Rh2 treatment, and four VEGFs were down-regulated by (20S)G-Rh2 (Figure 4A). HepG2-HUVEC co-culture was performed with (20S)G-Rh2-treated HepG2 cells. (20S)G-Rh2 relieved the pro-colony-formation capability of HepG2 CM on both colony counts and colony population (Figure 4B,C). The cell cycle of HUVECs was altered as G0/G1-phase cells increased and S/G2/M-phase cells decreased (Appendix A and Figure 2B). HUVECs co-culture with (20S)G-Rh2 treated HepG2 cells presented a less EdU positive ratio, presenting a weaker DNA synthesis and cell proliferation (Figure 4D,E). The pro-migration capability of HepG2 was inhibited by (20S)G-Rh2 in a trans-well assay for HUVECs (Figure 4F and Appendix A).

### 2.5. (20. S)G-Rh2 Interacts with Anxa2-Y23D Mutant

Anxa2-Tyr23 phosphorylation is a key modification for Anxa2-STAT3 interaction [23,24]. In order to further confirm the regulatory mechanism of (20S)G-Rh2 on STAT3 activation, thermal shift assay was performed with phosphomimic (Y23D) and non-phospho-mimic (Y23A) of Anxa2, and both Y23A and Y23D mutant interacted with (20S)G-Rh2 in HepG2 cells or as purified free proteins (Figure 5A,B). Anxa2-STAT3 interaction and STAT3 activation were determined in Anxa2-over-expressed HepG2 cells. Wild-type (WT) and Y23D mutant of Anxa2 showed strong interaction and pro-activation capability to STAT3 (Figure 5C–E). The expression of VEGF A-D was enhanced on both mRNA and protein levels (Figure 5F,G), and the secreted VEGFs were also enhanced (Figure 6A). As a consequence, the colony formation of HUVECs was promoted by Anxa2-WT and Anxa2-Y23D over-expressed HepG2 cells (Figure 6B,C). The cell cycle alteration of HUVECs presented a further decrease in G0/G1-phase and an addtional increase in S/G2/M-phase (Appendix A). HUVECs showed a higher EdU-positive ratio (Figure 6D,E), and the invasive capability of HUVECs was also enhanced (Figure 6F and Appendix A). Y23A-Anxa2, as a non-phospho-mimic, showed weaker interaction with STAT3 (Figure 5C) and weaker pro-activation capability towards STAT3 (Figure 5D,E). With no significantly increased expression and secretion of VEGFs (Figure 5F,G and Figure 6A), Y23A-Anxa2 over-expressed HepG2 presented no pro-proliferation and pro-invasion effect on HUVECs (Figure 6B–G, Appendix A). (20S)G-Rh2 interfered the interaction between STAT3 and Anxa2-WT/Y23D (Figure 5C), and inhibited the enhanced activation of STAT3 (Figure 5D,E) as well as down-stream VEGF expression (Figure 5F,G and Figure 6A). The proliferation and migration were also suppressed by (20S)G-Rh2 when HUVECs were co-cultured with Anxa2-WT/Y23D over-expressed HepG2 (Figure 6B–F and Appendix A).

### 2.6. K301A Mutant of Anxa2 Protects HepG2 Cells from (20S)G-Rh2 Induced STAT3 Inhibition

In our previous study, we have already described Lys301 of Anxa2 is a key amino acid residue for Anxa2-(20S)G-Rh2 interaction [27,28]. Here, we over-expressed Anxa2 with double site mutation on Tyr23 and Lys301 in HepG2 cells for further investigation on STAT3 activation. Thermal shift assay was firstly performed, and Anxa2 with K301A mutation failed to interact with (20S)G-Rh2 (Figure 7A,B). The K301A mutation showed no effect on STAT3 activation as that Anxa2-K301A(Anxa2-YA) and Anxa2-Y23D/K301A (Anxa2-DA) interacted with STAT3 and promoted Tyr705 phosphorylation of STAT3 and the following STAT3 activation (Figure 7C–E). VEGFs were up-regulated in Anxa2-YA and Anxa2-DA over-expressed HepG2 cells (Figure 7F,G and Figure 8A) with an enhancement on the growth and migration of co-cultured HUVECs (Figure 8B–F and Appendix A). On the other hand, (20S)G-Rh2 failed to interfere the interaction between STAT3 and Anxa2-YA or Anxa2-DA and following STAT3 activation (Figure 7C–E). Anxa2-YA or Anxa2-DA mutant in HepG2 cells maintained VEGF expression (Figure 7F,G and Figure 8A) and further pro-proliferation and pro-migration capability (Figure 8B–F and Appendix A).

## 3. Discussion

STAT3 transits extracellular cytokine stimuli to specific gene transcription facilitating cell growth, survival, motility and inflammatory response [1,2,3,4,5,6]. In tumor cells and tumor microenvironment, excessive cytokines and functional mutation of kinase pathway members alter the regulation of tyrosine kinase pathway, providing a hyper-phosphorylation status at STAT3- Tyr705, which drives the head-to-tail dimerization and following activation [35,36]. With hyper- and sustaining-activation in over 70% of human cancers, STAT3 is widely considered as an oncogene and therapeutic target in various types of cancers, and a series of chemicals or targeted biomacromolecules have been employed in clinical cancer treatment. Current targeted approaches towards STAT3 are mainly concentrated in the EGFR/Src/STAT3 and IL6/JAK/STAT3 pathway, including related tyrosine kinase inhibitor (TKI), receptor inhibitor, STAT3-SH2 domain inhibitor and anti-sense nucleotide targeting STAT3 [5,6,11,35]. For the complexity of the mutation variation and compensative access in tumor, the tumor type specificity including the mutation landscape has become a primary concern before the utilization of STAT3 inhibitors [35,37].

Within the process of STAT3 activation, Anxa2 with a phosphor-Tyr23 is identified as a novel dominant regulator for STAT3-Tyr705 phosphorylation, which is responsible for nuclear entry and transcriptional activation of STAT3 (Figure 1A–C). As Anxa2 expression and phosphor-Tyr23 are largely evaluated in various cancer types [16,17,18,19,20,21,34], this regulatory mechanism towards STAT3 somehow outstands as an assignable target for STAT3 inhibition. Anxa2 has been proven facilitating cancer progression via multiple aspects by interacting with other biomolecules, among which the Anxa2 rearrangement gene expression in cancer cells through oncogenic transcription factor like NF-κB, YAP1 in Hippo pathway and STAT3/6 [22,26]. In our previous research, (20S)G-Rh2 is indicated as a natural inhibitor for Anxa2 and depresses NF-κB activation via Anxa2-p50 subunit interaction [27,28]. Down-stream genes like IL-6, IAPs, and EMT transcription factors are narrowed under (20S)G-Rh2 treatment, resulting in the loss of EMT and anti-apoptosis capability [27,28].

In this article, we primarily obtained the down-regulation of four VEGFs along with STAT3 suppression in Anxa2-knockdown cells (Figure 1C–E). As VEGFs are key promoters for endothelial cell proliferation and migration, co-culture systems containing HepG2 cells and HUVECs were generated, and both proliferation and migration of HUVECs were enhanced co-culturing with HepG2 cells (Figure 2 and Appendix A). With the loss of Anxa2 expression and following STAT3 activation, HepG2 cells presented no promotive capability to HUVECs (Figure 2 and Appendix A). To value the targeted effect of (20S)G-Rh2, a thermal shift assay was performed and (20S)G-Rh2 interacted with both non-modified and phosphorylated Anxa2 (Figure 3A,B). The phospho-status of Anxa2 showed no alteration under (20S)G-Rh2 treatment while Anxa2-STAT3 interaction and STAT3 phosphorylation as well as activation were reduced, indicating (20S)G-Rh2 inhibited STAT3 activation via targeting Anxa2-STAT3 interaction but not post-translational modification of Anxa2. Four VEGFs were down-regulated under STAT3 inhibition by (20S)G-Rh2, and HepG2 cells with lower VEGF expression and secretion presented no promotion on the proliferation and migration of HUVECs (Figure 4 and Appendix A)

The variety of post-transcriptional modification is the key function switch for Anxa2-regulated cellular events, among which phospho-Tyr23 is essential for STAT3 activation [23,24]. As to view the insight of (20S)G-Rh2 regulated STAT3 inhibition, we committed the additional Anxa2 expression with mutation at Tyr23 and Lys301, which respectively controlled its binding to STAT3 and (20S)G-Rh2. Anxa2 with Y23D mutation or of wild type presented binding-capable to STAT3, while no binding capability was gained for Anxa2-Y23A mutant (Figure 5C and Figure 7C), and only binding-capable Anxa2 enhanced STAT3 activation and following proliferation- and migration- promotive effect on HUVECs (Figure 5D–F, Figure 6, Figure 7D–F, Figure 8, Appendix A). Moreover, mutation at Tyr23 showed no impact on Anxa2-(20S)G-Rh2 interaction, and (20S)G-Rh2 inhibited STAT3 activation in HepG2 cells expressing mutated Anxa2 at Tyr23 (Figure 5, Figure 6, and Appendix A). On the other hand, Anxa2-K301A mutant, a (20S)G-Rh2-binding-deficient mutant, remained its binding capability to STAT3 under (20S)G-Rh2 treatment and protected HepG2 cells from the inhibition of STAT3 by (20S)G-Rh2 (Figure 7, Figure 8 and Appendix A), demonstrating the targeted effect of (20S)G-Rh2 towards Anxa2-dependent STAT3 pathway.

(20S)G-Rh2 is a well-described natural chemical from ginseng for its anti-cancer activity, and part of targets and mechanisms has lately been revealed as (20S)G-Rh2 inhibited NF-κB by interacting with Anxa2 [27,28]. In this article, we expanded the targeted effect of (20S)G-Rh2, and systematically described its inhibition towards STAT3 and downstream VEGFs, suggesting the possibility that (20S)G-Rh2 interfered tumor micro-environment, providing new molecular basis and strategies for the further utilization of (20S)G-Rh2 in cancer therapy and research.

## 4. Materials and Methods

### 4.1. Cell Lines and Culture

Human liver cancer cell line HepG2 (HB-8065, ATCC, Manassas, VA, USA) and human umbilical vein/vascular endothelium cell line HUVEC (CRL-1730, ATCC, Manassas, VR, USA) were cultured in DMEM high glucose medium (Gibco, Grand Island, NY, USA) and Ham’s F-12K medium (Gibco, Grand Island, NY, USA) respectively supplemented with 10% fetal bovine serum (BI, Belt Haemek, Isreal), 100 units/mL penicillin and 100 μg/mL streptomycin, in a humidified 5% CO_2_ atmosphere at 37 °C.

### 4.2. Chemicals, Antibody, and Plasmids

(20S)G-Rh2 (Sigma-Aldrich, St. Louis, MO, USA) was dissolved in 75% ethyl alcohol to a final concentration of 10 mM. Saracatinib (MCE, Monmouth Junction, NJ, USA) was dissolved in DMSO to a final concentration of 1 mM. Src inhibitor 3 (MCE, Monmouth Junction, NJ, USA) was dissolved in DMSO to a final concentration of 1 mM.

Antibodies for Anxa2 (sc-47696), VEGFA (sc-7269), VEGFB (sc-80442), VEGFC (sc-374628), VEGFD (sc-373866), STAT3 (sc-8019), pY23-Anxa2 (sc-135753), c-Src (sc-8056) and GAPDH (sc-47724) were purchased from Santa Cruz Biotechnology (Dallas, TX, USA). Antibodies for pY705-STAT3 (AF3293) and pY23-Anxa2 (AF7096) were purchased from Affinity (Affinity Biosciences LTD., Cincinnati, OH, USA). Antibodies for myc-tag (60003-2-Ig and 16286-1-AP) were purchased from Proteintech (Proteintech Group Inc., Rosemont, IL, USA). HRP-conjugated goat anti-rabbit IgG (H + L) secondary antibody (31460) and HRP-conjugated goat anti-mouse IgG (H + L) secondary antibody (31430) were purchased from Invitrogen (Invitrogen, Grand Island, NY, USA).

The plasmids for wild -type and K301A mutated Anxa2 expression (pcs4-Anxa2-WT-myc, pcs4-Anxa2-K301A-myc, pEXS-Anxa2-WT and pEXS-Anxa2-K301A) were shown as described [27,28]. Y23A and Y23D mutation of Anxa2 were generated by point mutation within Anxa2-WT and Anxa2-K301A expressing vectors. Anxa2-shRNA vectors, pGPU6-GFP-Neo-Anxa2, were obtained from GenePhama (GenePhama, Suzhou, China), with sequence details shown in Appendix A. The plasmid for constitutively active c-Src (Y530F, Src-CA) was a gift from Jae Youl Cho (Department of Genetic Engineering, Sungkyunkwan University). Dual luciferase reporter assay were performed with pSTAT3-TA-luc (Beyotime, Shanghai, China) and pRL-CMV (Promega, Fitchburg, WI, USA).

### 4.3. Cell Co-Culture Systems

Three co-culture systems were utilized to determine the promotive effect of HepG2 cells towards vein/vascular endothelium cell.

For colony formation analysis and colon population counting: HepG2 cells pre-treated with chemical or plasmid transfection were cultured with serum-free F-12K medium for 24 h in 100-mm culture dish, and the supernatant was collected followed by a centrifugation of 12,000× *g* for 10 min. Then, the supernatant was filtered with 0.22-μm film for further use. 1 × 10^3^ HUVECs were seeded onto a 6-well plate and cultured with filtered HepG2 supernatant containing 10% FBS. The culture medium was refreshed every 48 h in this duration up to 14 days, after which cells were stained with 0.25% (*w*/*v*) crystal violet and photographed.

For cell invasion assay: 1 × 10^4^ HepG2 cells were first seeded onto 24-well plate and treated with chemical or plasmid transfection. The supernatant was then replaced with F-12K medium containing 20% FBS and applied as the lower chamber. Then, 1 × 10^4^ HUVECs were seeded in the upper chamber of 12-μm trans-well chamber (Corning, Corning, NY, USA) coated with Matrigel (356234, Corning, Corning, NY, USA) and cultured for 24 h followed by crystal violet staining and photographing.

For EdU staining and cell cycle analysis: 1 × 10^4^ HUVECs were seeded onto 24-well plate and applied as the lower chamber. Then, 1 × 10^4^ HepG2 cells were pre-treated with chemical or plasmid transfection and seeded in the upper chamber of a 3-μm trans-well chamber (Corning, Corning, NY, USA). Both cells were cultured for 48 h in serum-free F-12K medium and HUVECs were used for EdU labeling or cytometry analysis.

## 5. Immuno-Precipitation

50 μL of Protein A/G Magnetic Beads (K0202, MCE, Monmouth Junction, NJ, USA) was balanced with 400 μL of IP lysis buffer (Pierce, Rockford, IL, USA) for three times. 5 μg of antibody for immune-precipitation was diluted with IP lysis buffer and rotated with balanced beads for 2 h at 4 °C. HepG2 cells was collected and lysed with IP lysis buffer supplemented with Protease Inhibitor Cocktail (Roche, Indianapolis, IN, USA), Phospho-STOP Cocktail (MCE, Monmouth Junction, NJ, USA) and 1-mM phenylmethanesulfonyl fluoride (PMSF). Cell lysis with 500 μg of protein was diluted to the final volume of 400 μL and then combined with antibody-bonded beads, followed by another rotation for 2 h at 4 °C. The beads were then washed with IP lysis for three times and collected for immune-blot analysis.

## 6. Thermal Shift Assay

In vivo thermal shift assay (cellular thermal shift assay, CTSA): 3 × 10^7^ HepG2 cells were cultured with 20 μM (20S)G-Rh2 for 1 h at 37 °C and then resuspended with PBS containing 1 mM PMSF to a final density of 2 × 10^7^/mL. Cells were subpackaged into 12 PCR tubes with 100 μL per tube and then heated with a thermal gradient for 40 °C to 73 °C for 3 min. After freeze-thaw twice with liquid nitrogen, the supernatant was separated by a centrifugation at 20,000× *g* for 20 min. 20 μL of the supernatant was loaded for immune-blot analysis.

In vitro thermal shift assay: Anxa2 purification was performed as described [27]. GST-tagged Anxa2 was expressed in *E. coli* BL21(DE3) strain. After loaded onto GST affinity column, GST tag was removed by HRV 3C protease. The purified protein was finally collected after separation by Superdex75 16/600 and 10-Kd centrifuge filter. Purified Anxa2 protein was diluted with PBS containing 1 mM PMSF to a final concentration of 0.2 mM (~7 μg/mL) and subpackaged into PCR tube with 90 μL per tube. (20S)G-Rh2 was diluted with PBS and then combined with Anxa2 protein as a dose gradient from 0.2 μM to 10 μM. PCR tubes were then made up to 100 μL PBS and heated at 55 °C for 3 min, with the control tube on ice for 3 min. The supernatant was separated by a centrifugation at 20,000× *g* for 20 min, and 10 μL of each tube was loaded for immune-blot.

## 7. Dual Luciferase Reporter Assay

pSTAT3-TA-luc and pRL-CMV (10:1, *w*:*w*) were co-transfected into HepG2 cells for 24 h before chemical treatment. The activity of luciferase was determined with Dual-Luciferase^®^ Reporter Assay System (Promega, E1910) according to the manufacture’s protocol. Luminescence was collected via Infinite F200 Pro (TECAN, Männedorf, Switzerland).

## 8. Real-Time Polymerase Chain Reaction (qRT-PCR)

Whole-cell RNA was isolated with TRIzol (Invitrogen, Grand Island, NY, USA) and 2 μg whole-cell RNA was proceeded with High Capacity cDNA Reverse Transcription Kit (4368814, Applied Biosystems, Foster city, CA, USA) for cDNA synthesis. Real-time PCR analysis was performed with PowereUp SYBR Green Master Mix (A25742, Applied Biosystems, Foster city, CA, USA) and 7500 Real-time PCR system (Applied Biosystems, Foster city, CA, USA). Prime pairs involved were in Appendix A.

## 9. Enzyme-Linked Immune Sorbent Assay (ELISA)

HepG2 cells were culture with serum-free DMEM for 24 h after a treatment with chemical or under plasmid transfection. The supernatant was centrifuged at 12,000× *g* for 10 min and collected for ELISA. VEGFA, VEGFB, VEGFC, and VEGFD in the supernatant were determined by ELISA kit (BMS277-2, EH481RB, BMS297-2, EHFIGF, Invitrogen Invitrogen, Grand Island, NY, USA).

## 10. Statistical Analysis

All data were obtained from independent triple-replicated experiment and presented as the mean ± standard deviation (SD). Significance was determined by a two-tail Student’s test via SPSS v18.0.

## Figures and Tables

**Figure 1 ijms-22-09289-f001:**
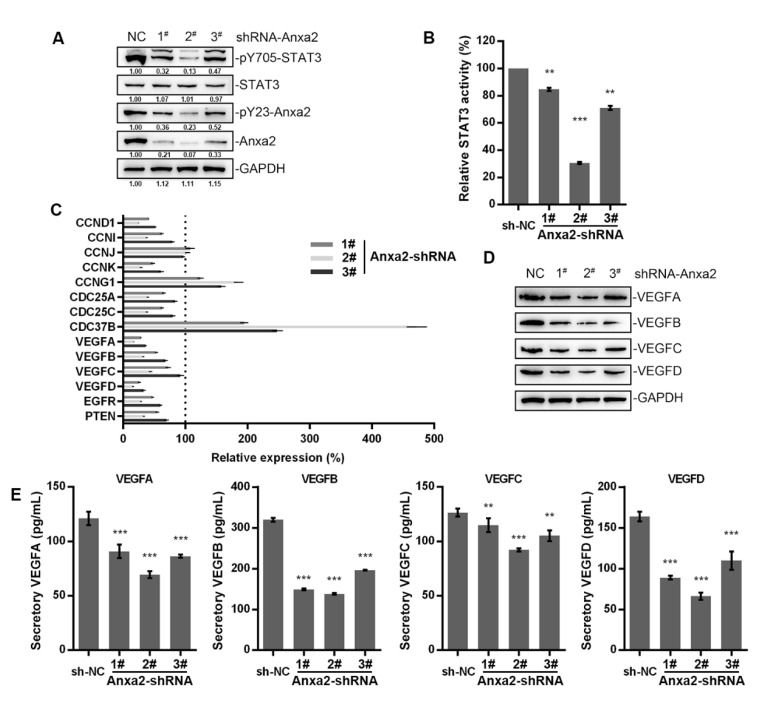
Anxa2 regulated the expression of VEGFs. HepG2 cells were treated with three different Anxa2-shRNAs. (**A**) Protein level and phosphorylation status of STAT3 and Anxa2 were determined by immuno-blotting. (**B**) The activation of STAT3 was determined via reporter gene assay. (**C**) The expression of STAT3 target genes was determined by qRT-PCR. (**D**) The protein level of four VEGFs was determined by immuno-blotting. (**E**) The concentration of four VEGFs in cell-free supernatant was determined by ELISA. All experiments were performed with triple replicant. Significance analysis was shown as ** (*p* < 0.01) and *** (*p* < 0.001).

**Figure 2 ijms-22-09289-f002:**
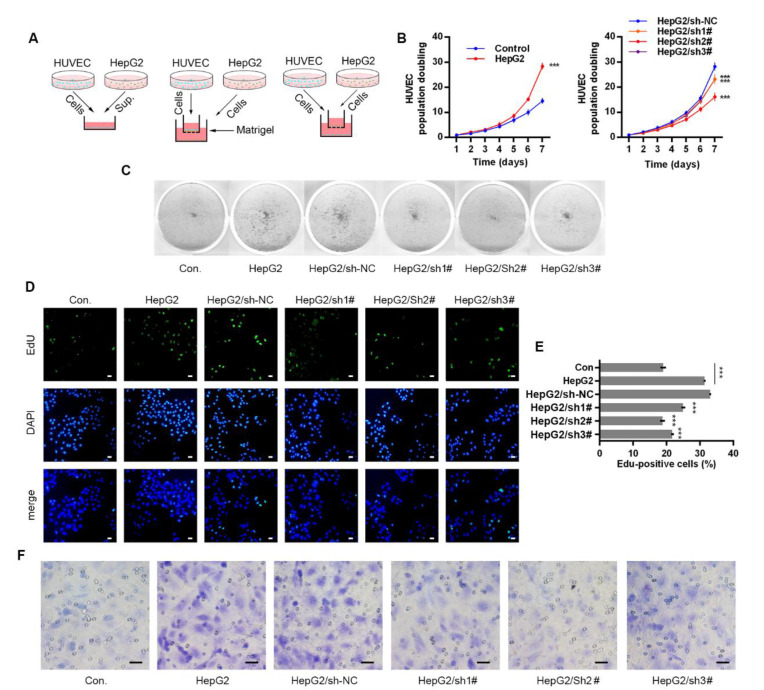
Anxa2 in HepG2 cells promoted proliferation and migration of co-cultured HUVECs. (**A**) Three co-culture arrangements were shown as schematic plots. (**B**,**C**) Colony formation of HUVECs were performed with cell-free supernatant from HepG2 cells with Anxa2 knock-down or not. The cell amount of each colony was shown as superimposed symbols with connecting lines (**B**), and the overall status of colony formation was shown as images after crystal violet staining (**C**). (**D**,**E**) EdU-labeling was utilized for new DNA synthesis detection in HUVECs co-cultured with HepG2 cells. EdU-positive cells were shown as florescent images (**D**), and EdU-positive ratio was shown as histogram (**E**,**F**) Migrated HUVECs co-cultured with HepG2 cells were shown as images after crystal violet staining. All experiments were performed with triple replicant. Significance analysis was shown as *** (*p* < 0.001). Scale bar presented 20 μm.

**Figure 3 ijms-22-09289-f003:**
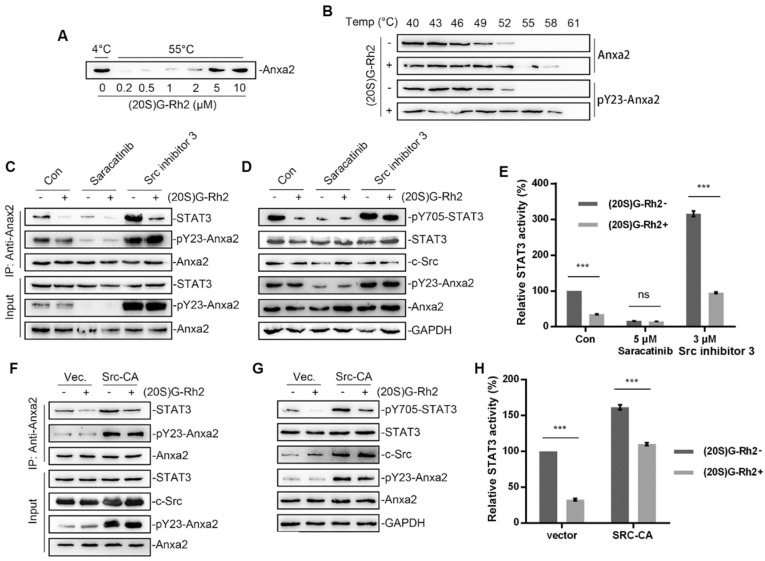
(20S)G-Rh2 inhibited STAT3 activation by targeting Anxa2. (**A**) Thermal shift assay was performed with purified Anxa2 and (20S)G-Rh2 gradient at 55 °C, and determined by immuno-blotting. (**B**) Cellular thermal shift assay was performed in HepG2 cells with indicated temperature gradient and determined by immuno-blotting. (**C**–**E**) HepG2 cells were treated with Src inhibitor (Saracatinib), Src activator (Src inhibitor 3) or (20S)G-Rh2, or under co-treatment. Immuno-precipitation with anti-Anxa2 antibody (**C**) and protein level (**D**) were determined by immuno-blotting. STAT3 activation was determined via reporter gene assay (**E**). (**F**–**H**) HepG2 cells were transfected with the constitutively activated mutant of Src and treated with (20S)G-Rh2. Immuno-precipitation with anti-Anxa2 antibody (**F**) and protein level (**G**) were determined by immuno-blotting. STAT3 activation was determined via reporter gene assay (**H**). All experiments were performed with triple replicant. Significance analysis was shown as ns (*p* > 0.05) and *** (*p* < 0.001).

**Figure 4 ijms-22-09289-f004:**
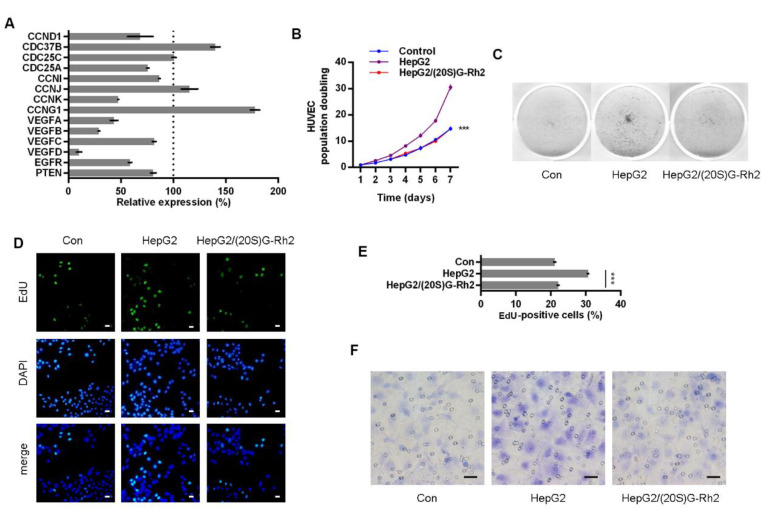
(20S)G-Rh2 inhibited HepG2 cells induced proliferation and migration of HUVECs. (**A**) Expression level of STAT3 target genes was determined in (20S)G-Rh2-treated HepG2 cells via qTR-PCR. (**B**, **C**) Colony formation of HUVECs were performed with cell-free supernatant from HepG2 cells pre-treated with (20S)G-Rh2 or not. The cell amount of each colony was shown as superimposed symbols with connecting lines (**B**), and the overall status of colony formation was shown as images after crystal violet staining (**C**). (**D, E**) EdU-labeling was utilized for new DNA synthesis detection in HUVECs co-cultured with HepG2 cells. EdU-positive cells were shown as florescent images (**D**), and EdU-positive ratio was shown as a histogram (**E**). (**F**) Migrated HUVECs co-cultured with HepG2 cells were shown as images after crystal violet staining. All experiments were performed with triple replicant. Significance analysis was shown as *** (*p* < 0.001). Scale bar presented 20 μm.

**Figure 5 ijms-22-09289-f005:**
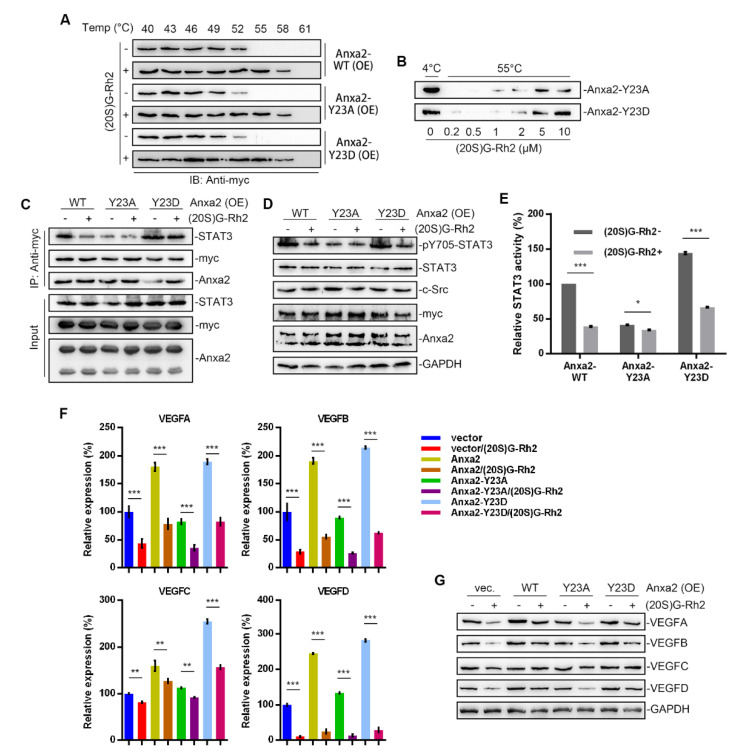
(20S)G-Rh2 interacted with Anxa2-Y23D mutant and inhibited related STAT3 activation. Myc-tagged wild-type, Y23A mutant, and Y23D mutant of Anxa2 was over-expressed in HepG2 cells. (**A**) Cellular thermal shift assay was performed with (20S)G-Rh2 at indicated temperature gradient in Anxa2-over-expressed HepG2 cells, and determined by immuno-blotting. (**B**) Thermal shift assay was performed with purified Anxa2 mutant and (20S)G-Rh2 gradient at 55 °C, and determined by immuno-blotting. (**C**–**E**) HepG2 cells with Anxa2 over-expression were treated with (20S)G-Rh2. Immuno-precipitation with anti-Anxa2 antibody (**C**) and protein level (**D**) were determined by immuno-blotting. STAT3 activation was determined via reporter gene assay (**E**). (**F**,**G**) Expression of four VEGFs in (20S)G-Rh2-treated HepG2 cell was determined by qRT-PCR (**F**) and immuno-blotting (**G**). Significance analysis was shown as * (*p* < 0.05), ** (*p* < 0.01) and *** (*p* < 0.001).

**Figure 6 ijms-22-09289-f006:**
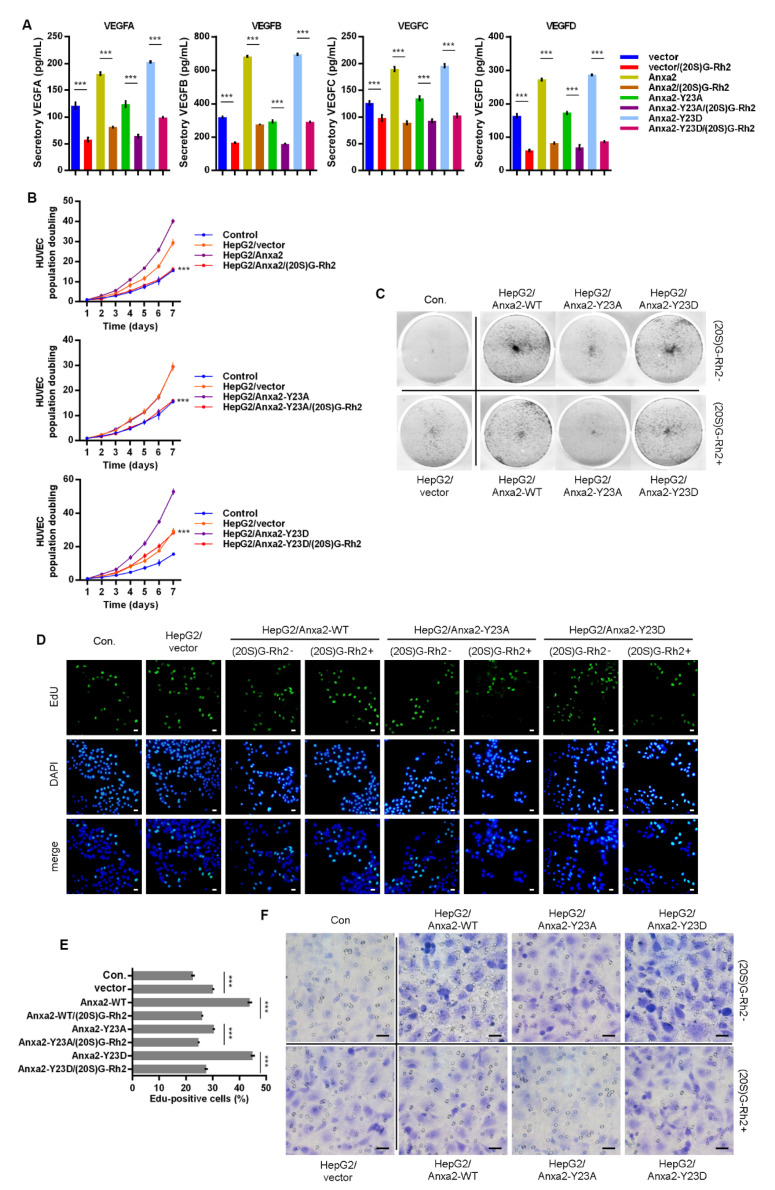
(20S)G-Rh2 inhibited Anxa2-Y23D mutant-enhanced proliferation and migration of HUVECs. Myc-tagged wild-type, Y23A mutant and Y23D mutant of Anxa2 was over-expressed in HepG2 cells. (**A**) The concentration of four VEGFs in cell-free supernatant of HepG2 cells was determined by ELISA. (**B**,**C**) Colony formation of HUVECs was performed with cell-free supernatant from HepG2 cells pre-treated with (20S)G-Rh2 or not. The cell amount of each colony was shown as superimposed symbols with connecting lines (**B**), and the overall status of colony formation was shown as images after crystal violet staining(**C**). (**D**,**E**) EdU-labeling was utilized for new DNA synthesis detection in HUVECs co-cultured with HepG2 cells. EdU-positive cells were shown as florescent images (**D**), and EdU-positive ratio was shown as histogram (**E**). (**F**) Migrated HUVECs co-cultured with HepG2 cells were shown as images after crystal violet staining. All experiments were performed with triple replicant. Significance analysis was shown *** (*p* < 0.001). Scale bar presented 20 μm.

**Figure 7 ijms-22-09289-f007:**
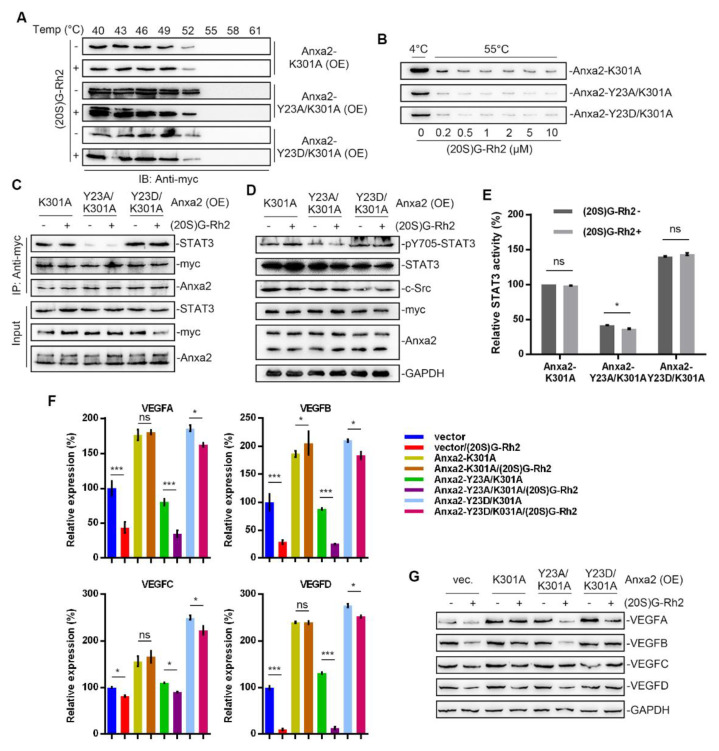
Anxa2-K301A mutant protected HepG2 cells from (20S)G-Rh2-induced STAT3 inhibition. Myc-tagged K301A, Y23A/K301A mutant and Y23D/K301A mutant of Anxa2 was over-expressed in HepG2 cells. (**A**) Cellular thermal shift assay was performed with (20S)G-Rh2 at the indicated temperature gradient in Anxa2-over-expressed HepG2 cells, and determined by immuno-blotting. (**B**) Thermal shift assay was performed with purified Anxa2 mutant and (20S)G-Rh2 gradient at 55 °C, and determined by immuno-blotting. (**C**–**E**) HepG2 cells with Anxa2 over-expression were treated with (20S)G-Rh2. Immuno-precipitation with anti-Anxa2 antibody (**C**) and protein level (**D**) were determined by immuno-blotting. STAT3 activation was determined via reporter gene assay (**E**). (**F**,**G**) Expression of four VEGFs in (20S)G-Rh2-treated HepG2 cell was determined by qRT-PCR (**F**) and immuno-blotting (**G**). Significance analysis was shown as * (*p* < 0.05) and *** (*p* < 0.001).

**Figure 8 ijms-22-09289-f008:**
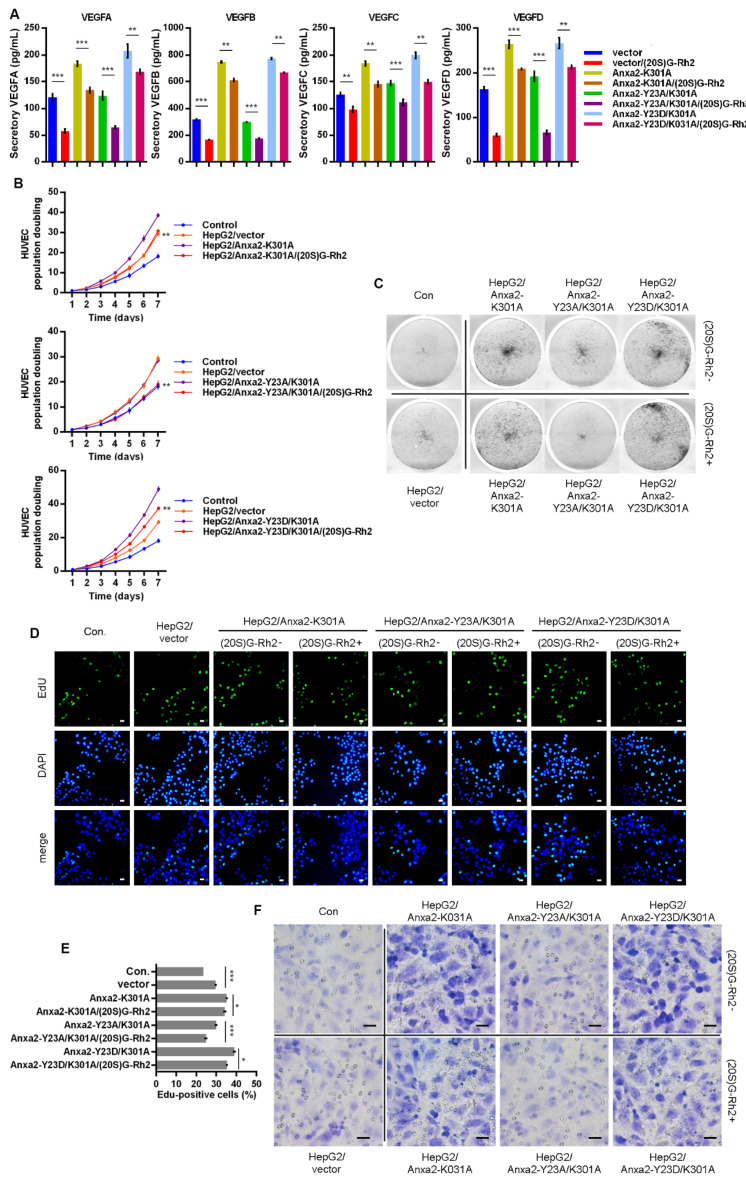
Anxa2-K301A mutant maintained pro-proliferation and pro-migration capability of HepG2 cells under (20S)G-Rh2 treatment. Myc-tagged K301A, Y23A/K301A mutant and Y23D/K301A mutant of Anxa2 was over-expressed in HepG2 cells. (**A**) The concentration of four VEGFs in cell-free supernatant of HepG2 cells was determined by ELISA. (**B**,**C**) Colony formation of HUVECs were performed with cell-free supernatant from HepG2 cells pre-treated with (20S)G-Rh2 or not. The cell amount of each colony was shown as superimposed symbols with connecting lines (**B**), and the overall status of colony formation was shown as images after crystal violet staining(**C**). (**D**,**E**) EdU-labeling was utilized for new DNA synthesis detection in HUVECs co-cultured with HepG2 cells. EdU-positive cells were shown as florescent images (**D**), and EdU-positive ratio was shown as a histogram (**E**). (**F**) Migrated HUVECs co-cultured with HepG2 cells were shown as images after crystal violet staining. All experiments were performed with triple replicant. Significance analysis was shown * (*p* < 0.05), ** (*p* < 0.01) and *** (*p* < 0.001). Scale bar presented 20 μm.

## Data Availability

Not available.

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
