# Peer review of "(20S) Ginsenoside Rh2 Inhibits STAT3/VEGF Signaling by Targeting Annexin A2"

_ijms, 2021, doi:10.3390/ijms22179289_

Round 1

Reviewer 1 Report

This manuscript entitled “(20S)G-Rh2 inhibits STAT3/VEGF signaling by targeting Annexin A2” is a follow-up study on the molecular targets of this natural compound in cancer cells.  Previous studies identified AnxA2 as the natural target for (20S)G-Rh2 and that this compound interfered with the interaction between NFkB and AnxA2. The present study describes yet another potential effect of (20S)G-Rh2 on another AnxA2 interacting protein STAT3. Here the authors propose that (20S) G-Rh2 interfered with the interaction between Annexin A2 and STAT3, and that this inhibited Tyr705 phosphorylation and STAT3 transcriptional activation. Inhibition of STAT3 activation by G-Rh2 inhibited the expression of VEGF ligands (and angiogenesis) as well as cell growth and motility. Overall the study lays the foundation for the use of (20S)G-Rh2 a natural inhibitor of Annexin A2 for therapeutic intervention against certain tumors via STAT3. While this support the notion that AnxA2 is a pro-angiogenesis factor, a number of inconsistencies need to be addressed.

  1. The expression of VEGFs is reported to be differentially affected by AnxA2 down regulation, mutant AnxA2 at Y23 and K301 as well as treatment of cells with G-Rh2. Based on these study, it not clear whether the growth and migration of HUVEC cells was due to activation of STAT3, phosphorylation of AnxA2 or expression and secretion of VEGFs. The authors should clearly describe their findings with this in mind.
  2. Cell cycle analysis of co-cultured HUVEC cells is presented in Figs 2, 4, 6, and 8. However, beyond indication that cell cycle analysis was performed, the implication of this analysis on the read-out including STAT3 activation, cell motility and/or cell growth needs to be discussed or eliminated from the manuscript.
  3. Src inhibition by Src inhibitor 3 and not Saracatinib (presumably very toxic) and Src-CA have similar effects on AnxA2 Y23 phosphorylation and STAT3 activation. G-Rh2 on the other hand, affected the interaction between Phospho-AnxA2 and STAT3 as well as the expression of STAT3 target genes. This suggests that another tyrosine kinase e.g. VEGF receptors rather than Src may be responsible for AnxA2 phosphorylation as well as G-Rh2 effects on VEGFs expression and HUVEC cell growth and migration.

Other issues:

There is a great deal of effort to present the data.  However, the manuscript as a whole requires extensive editing. This reviewer attempted to correct some of the language flaws, but could not continue.  A more logical presentation of the data is also warranted as it is highly redundant.

The title should contain the full name of the natural product “(20S) ginsenoside Rh2” rather than the current abbreviation.

Line 13, …interfered the interaction… should read …interfered with the interaction…

Line 27, Please provide a concrete example(s) of receptor-linked tyrosine kinase activated by cytokines, otherwise revise this section and in fact the first two paragraphs of the introduction since STATs are also activated by growth factors via receptor tyrosine kinases such as VEGF receptors.

The statement … predominantly activated by cytokines like EGF and IL-6… should read …predominantly activated by growth factors and cytokines like EGF and IL-6…via their cognate receptors.

Line 30, … genes containing cyclins…. Should read …genes including cyclins…

In the co-culture system (Figs. 3 and 4), it is not clear whether HUVEC cells were incubated with HepG2 cells with or without AnxA2 down regulation or were co-cultured in culture supernatants from HepG2 cells.  Authors should clearly indicate the experimental setup used in the co-culture and report the data as such in all the related figures.

For consistency use the same labeling for the AnxA2-shRNAs throughout the manuscript.

Figure 2H, 4H, 6G and 8G appear to vary only by the staining intensity and not by the number of migrated cells.  Please provide a quantitative analysis of all the migration assays.

Figure 3A and B, the authors should provide the rationale (basis for the temperature dependent variation of AnxA2 levels) for the thermal shift assay.

Fig. 3C-H. The Src inhibitors used and active Src did not substantially affect Y23-AnxA2 levels and Y705-STAT3 in HepG2 cells, while (20S)G-Rh2 affected total STAT3 interaction with AnxA2. This raises the question as to whether phosphorylation of AnxA2 is required for its interaction with STAT3 and/or activation of STAT3.

In lines 112/113, the statement C-Src inhibitors showed a widespread inhibition …. regulates the phosphorylation on Tyr705 of STAT3, should be supported by data and/or a prior studies (references).

For the relative activation in panels 1B, 3E, H 5E, etc., it is not clear what is referred to (AnxA2, STAT3?) in the Y-axis.

Lines 132-133, please indicate the cell lines in which this experiment was conducted (HepG2 or HUVECs).

Line 158, …(20S)G-Rh2 in vivo and in vitro should be changed to “in HepG2 cells and using purified proteins.

Lines 154-176 needs to be completely re-written as there is confusion as to whether the description refers to Fig.6 with uppercase or some other Figure 6 with lower case panels.

Lines, 121/122, 181, 218 please indicate whether the prokaryotic expressed AnxA2 was purified or not.  If purified, it suffice to simply state purified AnxA2.

Lines 331-340, three co-culture systems are indicated but only one of this system, the use of HepG2 spent media is described and may have been used.  Authors should clearly state the co-culture method(s) used and modify Fig. 2A accordingly.

Figs. 6A and 8A, the Y-axis refers to secreted VEGFs in pg/ml not “concentration”.

Author Response

Dear reviewer Thanks for your hard work and precious comments on this article. All questions and comments have been replied and related modification has been made in the newly subscribed manuscript and figures. Detailed explanation is shown in the attached file named point-to-point response Re1. Best regards.

Reviewer 2 Report

The article entitled ' (20S)G-Rh2 inhibits STAT3/VEGF signaling by targeting Annexin A2' is very interesting. The article describes the regulation of STAT3/VEGF signalling. The article can be improved with consolidated data.

  1. The authors nicely show the regulation of STAT3 by Anxa2. However, It is important to re-introduce wild type-Anxa2 in the shRNA- Anxa2 cells to see if the effects can be rescued.
  2. The authors show data from three shRNA towards Anxa2, can they comment which is the most efficient shRNA in the experiments.
  3. The data suggests Anxa2 and STAT3 are phosphorylated, however STAT3 levels are stable. Can they investigate if there are any changes at the transcriptional level by qPCR.
  4. In Fig1A, densitometry of the immunoblots are required.
  5. In fig 1B, they should correct as STAT3 relative quantification.
  6. Figure 2 H, the cells are very fussy. Can they show bright field pictures of it.
  7. It would be interesting to re-introduce Anxa2 in shRNA treated cells and see how the migrations is affected.
  8. Can they see localization of STAT3 upon its phosphorylation by Anxa2 by imaging?
  9. What happens to STAT3 upon phosphorylation is it translocated or degraded.
  10. Fig 6 D, the authors show cell cycle regulation, however there is not a significant difference between wild type and others. Can the authors comment about it.
  11. In Fig 7, Anxa2 has two bands which are very close or sometimes they are wide apart or there is a single band in multiple blots, can the authors justify the reason for it. It is confusing to know which one is the real band in the immunoblots.
  12. Can they show the specificity of the Anxa2 antibody that was used ?
  13. It is important to reintroduce AnXa2 in shRNA treated cells and probe for VEGF expression if it is rescued or not.
  14. An illustrative picture of the hypothesis is required to summarize the findings.

Author Response

Dear reviewer Thanks for your hard work and precious comments on this article. All questions and comments have been replied and related modification has been made in the newly subscribed manuscript and figures. Detailed explanation is shown in the attached file named point-to-point response Re2. Best regards.

Round 2

Reviewer 1 Report

In the attempt to respond to the critiques of the first review, the authors’ responses to certain critiques are unacceptable. A justification rather than defense of the data is required as this improves the quality of the work.

Regarding previous Q2 on cell cycle analysis, this reviewer is aware that cell cycle analysis indicates the distribution of percent of cells at different stages of the cell cycle. As provided in this manuscript, this analysis does not add any information to the data. Therefore, the authors should provide a rationale for this analysis and justify the differences on the read-outs of the study viz, expression/secretion of VEGFs, motility and/or growth of HUVEC cells.  Otherwise, all the cell cycle analyses should be moved to supplementary data.

On the Src inhibitors in the previous Q3, the response suggests that a Src related kinase Csk, rather than c-Src was targeted by one of the inhibitors.  There was no response on saracatinib and its potential cytotoxicity. It is therefore, not clear why Csk is targeted in Fig. 3C and D but c-Src-CA rather than Csk-CA is assessed in Fig. 3F and G. Although it appears that AnxA2 phosphorylation is mediated by Src family kinases (SFKs), it is unlikely that most of the members of this family of tyrosine kinases phosphorylate AnxA2 at Y23.  The authors should provide a solid justification for this analysis or provide data and/or references in which more broad spectrum SFK inhibitors were used.

Author Response

Dear reviewer

Thanks for your hard work and precious comments on this article. 

It appears to be not sufficient for cell cycle analysis to describe cell proliferation, and it should be appropriate to move cell cycle related data to supplementary files. This change has been made in the revised version.

For previous Q3, Saracatinib is a specific c-Src kinase inhibitor (not for Src kinase family, ), and presents inhibition towards c-Src, Lck, c-YES, Fyn, Fgr and Blk (official web site of MedChemExpress, MCE), which dose not cover the kinase responsible for STATA3-Tyr705 (JAKs, PTK6, FER). Src inhibitor 3 is a specific inhibitor for CSK (c-Src C-terminal kinase). CSK phosphorylates c-Src at Tyr530, which directly induces c-Src inhibition. So Src inhibitor 3 can be regarded as a c-Src activator.  On the other hand, c-Src is responsible for Anxa2-Tyr23 phosphorylation, which is not related to Src kinase family. All these information is justified as required in specifications for chemical or in references. On this point of view, it can be concluded as 'Saracatinib (c-Src inhibitor) inhibits pY23-Anxa2', 'Src inhibitor 3 (CSK inhibitor functioning as c-Src activator) promotes pY23-Anxa2' and 'Src-CA(additional active Src) promotes pY23-Anxa2'. All these conditions are aimed at provide different pY23-Anxa2 status for the determination of STAT3-Anxa2 binding inhibition by (20S)G-Rh2.  An additional description for Saracatinib and Src inhibitor 3 has been added in the revised version.

Best regards

Reviewer 2 Report

The article is interesting and the graphic abstract is really good along with the article. 

Author Response

Dear reviewer

Thanks for your comments and praise for this manuscript.

Couples of language modification have been made in the revised version.

Best regards.